# Munc18-1 is a dynamically regulated PKC target during short-term enhancement of transmitter release

Özgür Genç[1], Olexiy Kochubey[1], Ruud F Toonen[2], Matthijs Verhage[2,3], Ralf Schneggenburger[1]*

[1]Laboratory of Synaptic Mechanisms, Brain Mind Institute, School of Life Science, École Polytechnique Fédérale de Lausanne (EPFL), Lausanne, Switzerland; [2]Department of Functional Genomics, Center for Neurogenomics and Cognitive Research, Neuroscience Campus Amsterdam, Amsterdam, Netherlands; [3]Department of Genetics, VU University Medical Center, VU University, Amsterdam, Netherlands

**Abstract** Transmitter release at synapses is regulated by preceding neuronal activity, which can give rise to short-term enhancement of release like post-tetanic potentiation (PTP). Diacylglycerol (DAG) and Protein-kinase C (PKC) signaling in the nerve terminal have been widely implicated in the short-term modulation of transmitter release, but the target protein of PKC phosphorylation during short-term enhancement has remained unknown. Here, we use a gene-replacement strategy at the calyx of Held, a large CNS model synapse that expresses robust PTP, to study the molecular mechanisms of PTP. We find that two PKC phosphorylation sites of Munc18-1 are critically important for PTP, which identifies the presynaptic target protein for the action of PKC during PTP. Pharmacological experiments show that a phosphatase normally limits the duration of PTP, and that PTP is initiated by the action of a 'conventional' PKC isoform. Thus, a dynamic PKC phosphorylation/de-phosphorylation cycle of Munc18-1 drives short-term enhancement of transmitter release during PTP.

**\*For correspondence:** ralf.schneggenburger@epfl.ch

**Competing interests:** The author declares that no competing interests exist.

## Introduction

Vesicle fusion and transmitter release at synapses is a fundamentally important signaling mechanism that guarantees fast information transfer between neurons (*Südhof, 2004*). Interestingly, transmitter release is not static, but the amount of quanta released with each presynaptic action potential (AP) can vary dynamically, depending on the recent history of presynaptic activity. During high-frequency activity, short-term depression leads to a decrease of release in many synapses, but following recovery from depression, transmitter release can overshoot, giving rise to short-term enhancement of release. Various forms of short-term enhancement, like facilitation, augmentation and post-tetanic potentiation (PTP), have been observed and discriminated based on their duration (*Magleby and Zengel, 1975*; *Zucker and Regehr, 2002*). There is good evidence that the longer-lasting of these, augmentation and PTP, depend on the activation of intracellular second messengers in the nerve terminal, which in turn influence the release machinery.

A second messenger with a centrally important role in presynaptic plasticity is diacylglycerol (DAG), which can activate both protein kinase-C and Munc13-1 (*Lackner et al., 1999*; *Rhee et al., 2002*; *Wierda et al., 2007*). During augmentation, activation of the presynaptic protein Munc13-1 either by DAG or by Ca$^{2+}$/Calmodulin results in increased vesicle priming (*Rosenmund et al., 2002*; *Junge et al., 2004*). The other common form of short-term enhancement, PTP, has been observed at excitatory synapses in hippocampus (*McNaughton, 1982*; *Lee et al., 2007*), cerebellum (*Beierlein et al., 2007*),

**eLife digest** Brain function depends on the rapid transfer of information from one brain cell to the next at junctions known as synapses. Small packages called vesicles play an important role in this process. The arrival of an electrical action potential at the nerve terminal of the first cell causes some vesicles in the cell to fuse with the cell membrane, and this leads to the neurotransmitters inside the vesicles being released into the synapse. The neurotransmitters then bind to receptors on the second cell, which leads to an electrical signal in the second cell. A protein called Munc18-1 has a central role in the fusion of the vesicle at the cell membrane.

The strength of a synapse—that is, how easily the first brain cell can impact the electrical behaviour of the second—can change, and this 'synaptic plasticity' is thought to underlie learning and memory. Long-term changes in synaptic strength require additional receptors to be inserted into the membrane of the second cell. However, synapses can also be temporarily strengthened: the arrival of a burst of action potentials—a tetanus—causes some synapses to increase the amount of neurotransmitter they release in response to any subsequent, single, action potential.

This temporary increase in synaptic strength, which is known as post-tetanic potentiation, requires an enzyme called protein kinase C; the role of this enzyme is to phosphorylate specific target proteins (i.e., to add phosphate groups to them). Now, Genç et al. have genetically modified a mouse synapse in vivo and shown that protein kinase C brings about post-tetanic potentiation by phosphorylating Munc18-1. Furthermore, pharmacological experiments show that proteins called phosphatases, which de-phosphorylate proteins, normally terminate the post-tetanic potentiation after about one minute. Taken together, the study identifies a target protein which is phosphorylated by protein kinase C during post-tetanic potentiation. The study also suggests that in addition to its fundamental role in vesicle fusion, the phosphorylation state of Munc18-1 can change the probability of vesicle fusion in a more subtle way, thereby contributing to synaptic plasticity.

calyx of Held brainstem synapses (*Habets and Borst, 2005*; *Korogod et al., 2005*) and at the neuromuscular junction (*Magleby and Zengel, 1975*). PTP has been shown to be sensitive to pharmacological inhibition of PKC (*Alle et al., 2001*; *Brager et al., 2003*; *Korogod et al., 2007*), and deletion of the PKC α and β genes in mice suppressed PTP at the calyx synapse (*Fioravante et al., 2011*). Therefore, PTP is likely caused by a presynaptic PKC phosphorylation step, but the target protein of PKC during PTP has remained unknown.

Munc18-1 is a member of the Sec1/Munc18 family of proteins essential for membrane fusion from yeast to mammals (*Verhage et al., 2000*; *Südhof and Rothman, 2009*). Munc18-1 has two consensus sites for PKC phosphorylation (*Fujita et al., 1996*) which become phosphorylated during depolarization of synaptosomes (*de Vries et al., 2000*), and which are necessary for phorbol ester potentiation of transmitter release in cultured neurons (*Wierda et al., 2007*). Therefore, Munc18-1 is a candidate for PKC phosphorylation during PTP. Nevertheless, there are other presynaptic proteins which might act as PKC targets. First, SNAP-25 and Synaptotagmin-1 (Syt1) have PKC consensus sites (*Shimazaki et al., 1996*, *Hilfiker et al., 1999*). However, no evidence was found for an involvement of SNAP-25 in the phorbol ester potentiation of synaptic transmission in hippocampal neurons (*Finley et al., 2003*), and the PKC/CaM kinase consensus site in Syt1 is not conserved in Syt2 (*Nagy et al., 2006*). Since robust PTP is also observed at synapses which express Syt2 as their main $Ca^{2+}$ sensor, like the calyx of Held and neuromuscular synapses (*Pang et al., 2006*), Syt1 is unlikely a general phosphorylation target for the induction of PTP. Second, ion channels like voltage-gated $K^+$ channels and $Ca^{2+}$ channels are targets of PKC phosphorylation (*Zamponi et al., 1997*; *Song et al., 2005*). However, studies at the calyx synapse have shown only marginal changes in the presynaptic AP waveform or presynaptic $Ca^{2+}$ influx during PTP (*Habets and Borst, 2006*; *Korogod et al., 2007*), or during phorbol ester mediated potentiation of transmitter release (*Lou et al., 2005*, *2008*). These findings argue against a major role of ion channel phosphorylation during PKC-dependent short-term enhancement of release. Therefore, it is attractive to hypothesize that a protein of the release machinery is a PKC target during short-term enhancement, and Munc18-1 is an interesting candidate for this role.

Here, we wished to study the role of Munc18-1 phosphorylation during PTP. Since PTP has been observed mainly at synapses of acute preparations but not in cultured synapses, we developed an in

vivo gene replacement strategy at the calyx of Held synapse, a large CNS model synapse at which PTP measurements are well established (see references above). We used a floxed mouse line in which exon 2 of the Munc18-1 coding gene *Stxbp1* is flanked by loxP sites (called *Munc18-1^{lox/lox}* mice; *Heeroma et al., 2004*), combined with in vivo virus-mediated protein expression (*Wimmer et al., 2004*) to recombine the floxed allele, and to re-express mutant or wild-type Munc18-1 protein. Using these approaches, we show that a transient PKC phosphorylation of Munc18-1 causes the increased trans-mitter release that underlies PTP. These results identify Munc18-1 as a PKC target protein during PTP, and suggest that Munc18-1, besides its essential role in catalyzing membrane fusion, can mediate a second-messenger modulation of the release machinery during presynaptic plasticity.

## Results

### A phosphatase determines the duration of PTP

Previous studies have found evidence for a role of PKC during PTP, a form of short-term enhancement of release (*Alle et al., 2001*; *Brager et al., 2003*; *Korogod et al., 2007*; *Fioravante et al., 2011*). However, it remains possible that the requirement for PKC merely represents a background PKC activity *permissive* for the induction of PTP (see discussion in *Korogod et al., 2007*). We hypothesized that if PTP is caused by a dynamic phosphorylation/de-phosphorylation cycle of a presynaptic protein, phos-phatase blockers should prolong the duration of PTP.

We studied PTP at the calyx of Held synapse in a slice preparation, by first testing baseline synaptic strength with double stimuli (interval, 10 ms) repeated every 10 s. PTP was induced every 5–7 min using 4 s 100 Hz trains of afferent fiber stimuli (*Figure 1A*, arrowheads). PTP induction trains under control conditions led to ~twofold PTP which decayed nearly completely over the next 3 min, similarly as shown previously (*Korogod et al., 2005*). Acute application of calyculin (1 μM), an inhibitor of phosphatases PP1 and PP2A (*Ishihara et al., 1989*), strongly prolonged the decay of PTP (*Figure 1A*). We estimated the decay rate of PTP by line fits (*Figure 1A*, grey and red line), and found that the PTP decay rate was slowed from $16.0 \pm 2.6\%/min$ to $4.04 \pm 1.7\%/min$ (*Figure 1B*; n = 7 cells; p<0.01). Calyculin acted without changing the baseline synaptic strength (*Figure 1A,C*; p=0.96), nor the peak PTP amplitude ($219 \pm 17\%$ and $227 \pm 17\%$, in control and calyculin respectively; p=0.7). Following removal of calyculin, PTP gradually reversed to its normal decay kinetics (*Figure 1A*).

It is conceivable that pharmacological block of phosphatases interferes with postsynaptic plasticity mechanisms, and thereby causes an apparent prolongation of PTP. To distinguish between pre- and postsy-naptic sites of action of calyculin, we first analyzed the paired-pulse ratio ($EPSC_2/EPSC_1$). Under control conditions, the paired-pulse ratio was decreased during the peak of PTP, and then recovered when PTP decayed back to baseline (*Figure 1D*, black data points). The decreased paired-pulse ratio confirms the view that PTP is presynaptic in origin, and mediated by an increased transmitter release (*Habets and Borst, 2005*; *Korogod et al., 2005*). In the presence of calyculin both the decay of PTP as well as the recovery of the paired pulse ratio were slowed, revealing a parallel regulation of both the synaptic strength and the paired-pulse ratio by calyculin (*Figure 1D*; red data points; n = 7 cells). Thus, in the presence of a phosphatase inhibitor, the increased transmitter release during PTP decays more slowly.

We further analyzed the quantal mechanism of PTP in the absence and presence of calyculin by analyzing the spontaneous (miniature) EPSCs (mEPSCs), sampled in 10 s intervals in-between evoked EPSCs. *Figure 1E* shows plots of the evoked EPSC amplitude (*top*), and of the individual mEPSC amplitudes and time-averaged mEPSC frequency (*bottom*), both in control conditions, and following application of calyculin (1 μM). Following the PTP induction trains, we found that the mEPSC frequency was increased and then relaxed back to baseline value over several tens of seconds, as shown before (*Korogod et al., 2005*). Under control conditions, the mEPSC amplitude was increased during the first 10 s interval following PTP induction (by 23% on average; *Figure 1F*; p<0.05). However, the increase in mEPSC amplitudes did not reach statistical significance in the other data sets of this study (see below, *Figure 2—figure supplement 1*, *Figure 4—figure supplement 1*). These findings are consistent with the view that PTP largely represents an increase in the amount of released quanta (*Korogod et al., 2005*), and that a significant increase in mEPSC amplitude, maybe caused by compound fusion, is only observed with stronger PTP induction stimuli (*He et al., 2009*; *Xue and Wu, 2010*). Interestingly, in the presence of calyculin, the elevated mEPSC frequency following PTP induction trains did not recover completely, but rather, persisted at 2–3-fold elevated levels (*Figure 1E*, bottom). This effect was observed in all cells in which mEPSC frequency was analyzed (n = 5; p<0.001; *Figure 1G*). Thus, the

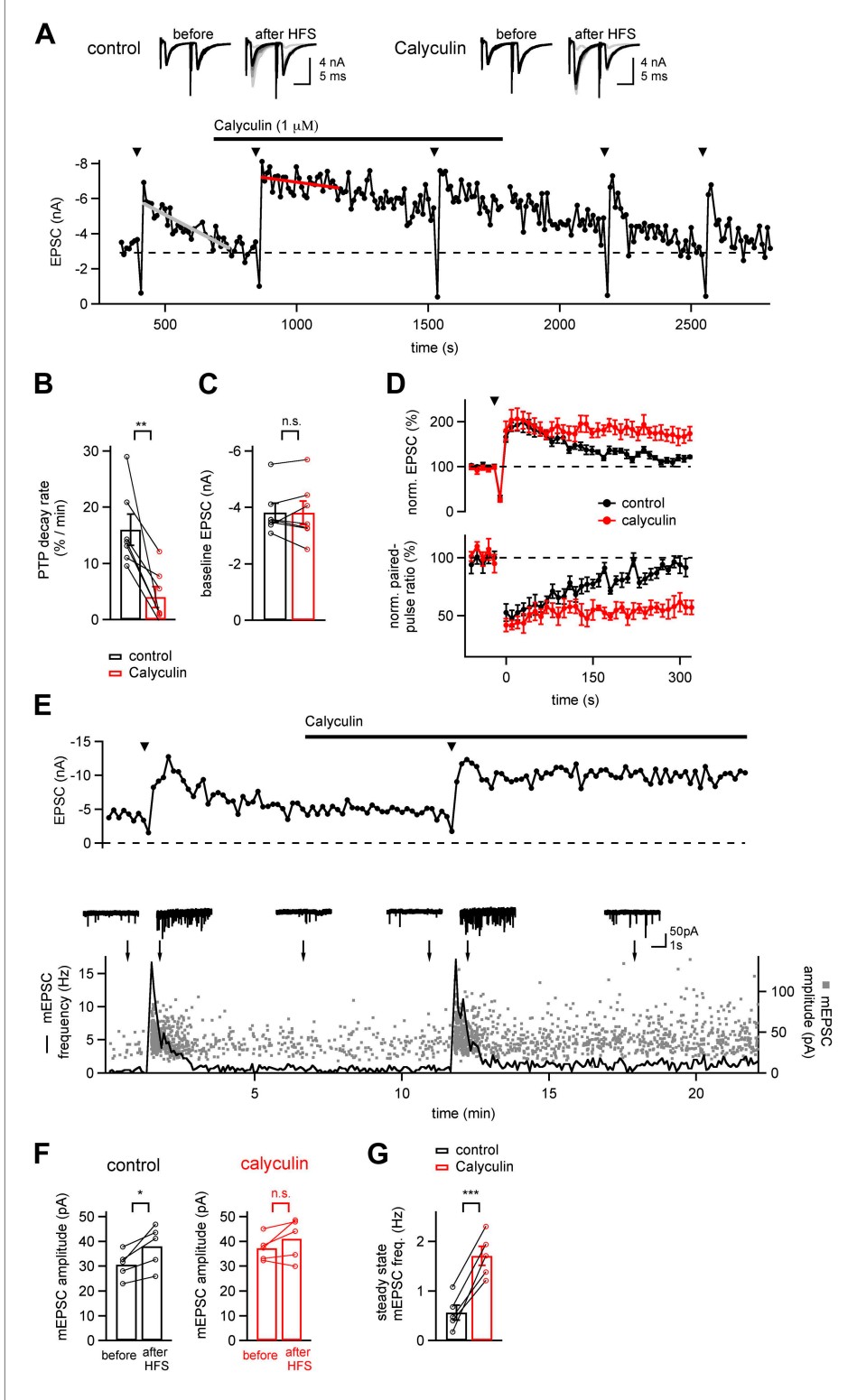

**Figure 1**. A phosphatase terminates the increased transmitter release which underlies PTP. (**A**) Time course of EPSC amplitude during the repetitive inductions of PTP (arrowheads, HFS at 100 Hz for 4 s), demonstrating the slowing of PTP decay upon acute application of phosphatase inhibitor Calyculin A. Line fits (grey and red lines) were used to estimate the decay rates of PTP. Upper inset shows example EPSCs induced by double stimuli (interval, 10 ms) before and after HFS, both for control condition (*left*) and following calyculin application (*right*).
*Figure 1. Continued on next page*

*Figure 1. Continued*

(**B** and **C**) Quantifications of PTP decay rates for control and calyculin (**B** and *Figure 1—source data 1A*) and basal EPSC amplitudes for the two conditions (**C** and *Figure 1—source data 1B*). (**D**) Average time courses of normalized EPSC amplitudes (*top*, PTP) and normalized paired-pulse ratio (EPSC$_2$/EPSC$_1$, *bottom*) in control conditions (black symbols) and in the presence of Calyculin A (red symbols), obtained in the same recordings (n = 7 cells). (**E**) Time course of evoked EPSCs (*top*) and mEPSC frequency (*bottom*, line trace) and scatter plot of individual mEPSC amplitudes (*bottom*, gray data points) as a function of experiment time, from a different example recording as the one shown in (**A**). Example mEPSC traces are shown for the time points indicated by arrows. (**F**) Quantification of average mEPSC amplitudes before PTP induction stimuli (sampled from at least five 10 s long mEPSC traces, left bars), and during a single 10 s interval immediately following PTP induction (right bars). Data for both control conditions (left) and in the presence of calyculin (right) are shown (see *Figure 1—source data 1C*). (**G**) Quantification of the mEPSC frequency late after induction of PTP, both under control conditions, and in the subsequent presence of calyculin (1 µM) in the same cell. Note that in the presence of calyculin, the steady state mEPSC frequency was persistently increased (see *Figure1—source data 1D*).

The following source data are available for figure 1:

**Source data 1**.

persistently enhanced transmitter release following PTP induction trains under phosphatase block is mirrored by an enhanced spontaneous release rate.

## Pharmacological evidence for a role of conventional PKCs

Phosphatases reverse the action of many serine–threonine kinases (*Hunter, 1995*). We therefore wanted to find further evidence that the action of a phosphatase is related to an initial PKC phosphorylation step. Furthermore, we wanted to distinguish pharmacologically whether 'novel' or 'conventional' PKCs initiate PTP (*Newton, 2001*). For this purpose, we made use of cell-permeable inhibitory peptides directed either against the conventional PKC isoforms α and βI–II (called PKCi), or against protein kinase A in a control experiment (called PKAi). We measured PTP in slices pre-incubated either with PKAi or with PKCi at 10 µM each (*Figure 2A*; see 'Materials and methods' for procedures of drug application). PKCi strongly suppressed PTP (*Figure 2A,B*; p<0.001), whereas PTP was normal in the presence of PKAi (*Figure 2A,B*). The baseline synaptic strength was similar under both conditions (*Figure 2C*; p=0.92). Since large EPSCs usually show smaller (relative) PTP at the calyx synapse (*Korogod et al., 2005*), we plotted the amount of PTP vs EPSC amplitude to control for a possible sampling bias with respect to baseline EPSC amplitudes. This revealed that PKCi also reduced PTP in recordings with small baseline EPSCs (*Figure 2D*). Therefore, the smaller average PTP in the presence of PKCi was not caused by a sampling bias towards large baseline EPSCs.

The experiments with the PKCi inhibitory peptide suggest that conventional PKCs (PKCα and -β) are involved in PTP at the calyx synapse, in agreement with recent genetic evidence (*Fioravante et al., 2011*). An earlier study had shown, however, that the novel PKC isoform PKCε becomes translocated in the calyx of Held nerve terminal upon phorbol ester stimulation (*Saitoh et al., 2001*). If novel PKCs are involved in PTP, one might expect a contribution of upstream phospholipase C signaling which produces DAG, since novel PKCs are activated by DAG but not by Ca$^{2+}$ (*Newton, 2001*). Therefore, we tested the role of the phospholipase C inhibitor U73122 (3 µM), but we could not find significant effects on PTP (*Figure 2E–G*). Similarly, another PLC inhibitor Neomycin (10 µM) did not suppress PTP (*Figure 2—figure supplement 2*), despite an immediate effect of Neomycin on transmitter release, which was probably caused by inhibition of presynaptic P/Q-type Ca$^{2+}$ channels (*Pichler et al., 1996*).

Taken together, the experiments with the phosphatase inhibitor calyculin and with the PKC inhibitory peptide strongly suggest that a dynamic phosphorylation/de-phosphorylation cycle, initiated by conventional PKC isoforms, determines the time course of PTP at the calyx of Held. PTP was insensitive to PLC blockers, consistent with the view that conventional, but not novel PKCs initiate the phosphorylation of a presynaptic target protein during PTP.

## Gene replacement of Munc18-1 at the calyx of Held synapse

We next wished to study the role of Munc18-1 phosphorylation for PTP at the calyx of Held. Munc18-1 is an essential protein for vesicle fusion and membrane trafficking (*Südhof and Rothman, 2009*),

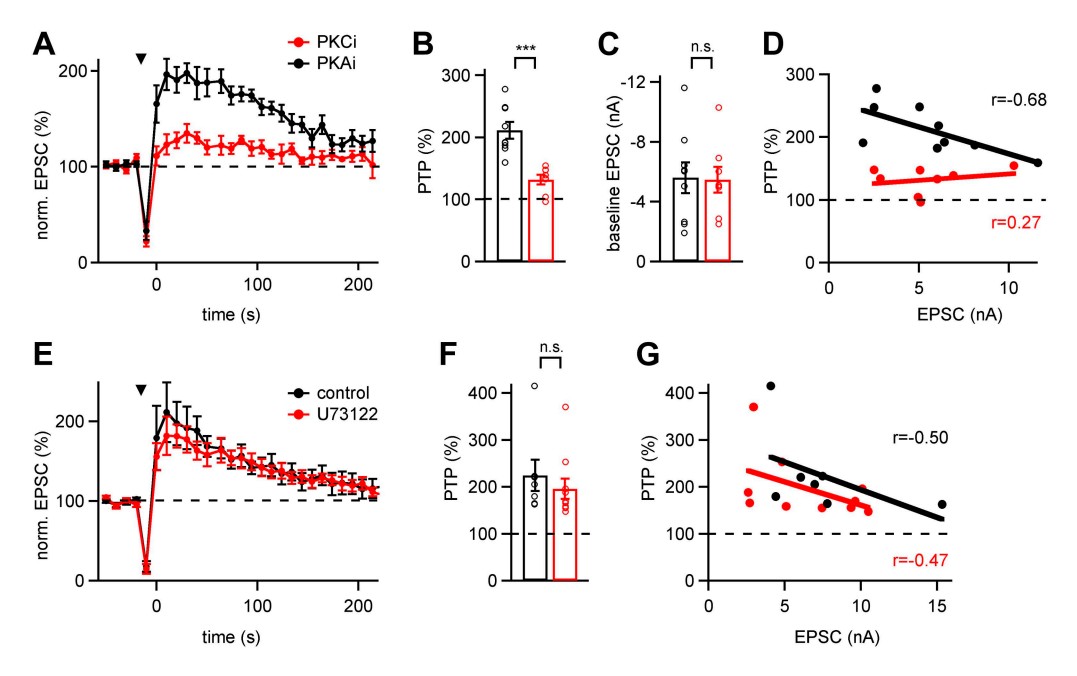

**Figure 2**. Conventional PKCs but not phospholipase-C initiate PTP. (**A**) Average time courses of normalized EPSC amplitude (PTP plots) in the presence of PKC inhibitory peptide (PKCi; red symbols, n = 8 cells) or PKA inhibitory peptide (PKAi; black symbols, n = 9 cells). Note the significant suppression of PTP by PKCi. (**B** and **C**) Quantifications of peak PTP (**B** and *Figure 2—source data 1A*) and of baseline EPSC amplitude (**C** and *Figure 2—source data 1B*) in the presence of PKCi (*red* symbols) and PKAi (*black*). (**D**) Plot of peak PTP amplitudes vs the basal EPSC amplitudes. Note that PTP was strongly reduced also when baseline EPSC amplitude was small. The correlation coefficients (r) are indicated. (**E**) Average PTP plots in neurons recorded after pre-incubation with the PLC blocker U73122 (red symbols; n = 10 cells), and under control conditions with 0.1% DMSO (black symbols; n = 7 cells). (**F** and **G**) Quantification of average peak PTP under control conditions and in the presence of U73122 (**F**, and *Figure 2—source data 1C*), and plot of peak PTP amplitude vs baseline EPSC amplitude (**G**, black and red symbols, respectively). Note the absence of an effect of phospholipase-C inhibition on PTP.

The following source data and figure supplements are available for figure 2:

**Source data 1**.

**Figure supplement 1**. mEPSC amplitude before and after PTP induction protocols is unchanged for the PTP data sets in *Figure 2*.

**Figure supplement 2**. PTP is insensitive to the PLC inhibitor Neomycin.

---

which has two PKC phosphorylation sites (*Fujita et al., 1996*; *de Vries et al., 2000*; *Wierda et al., 2007*). To investigate the role of Munc18-1 in PTP, it was necessary to develop a gene replacement strategy in which endogenous Munc18-1 protein is replaced with a PKC-insensitive mutant. Since constitutive genetic deletion of Munc18-1 in mice is lethal at the late embryo stage (*Verhage et al., 2000*), we used *Munc18-1^{lox/lox}* mice (*Heeroma et al., 2004*) and recombined the loxP sites using virus-mediated Cre expression. In addition, we wished to re-express phosphorylation-deficient Munc18-1, or wild-type Munc18-1 using the same viral construct. This required the simultaneous expression of three proteins: (i) Munc18-1, in the wild-type or mutant form, (ii) Cre-recombinase, and (iii) GFP to label successfully transduced calyces of Held. We used an adenovirus vector which allows the use of two independent expression cassettes (*Young and Neher, 2009*); one of these carried an internal ribosome entry site (IRES) to allow the expression of a third protein (*Figure 3A3*). Several additional constructs were prepared for control experiments (*Figure 3A1,A2*). We injected the adenovirus into the ventral cochlear nucleus (VCN) of *Munc18-1^{lox/lox}* mice early postnatally, to allow for

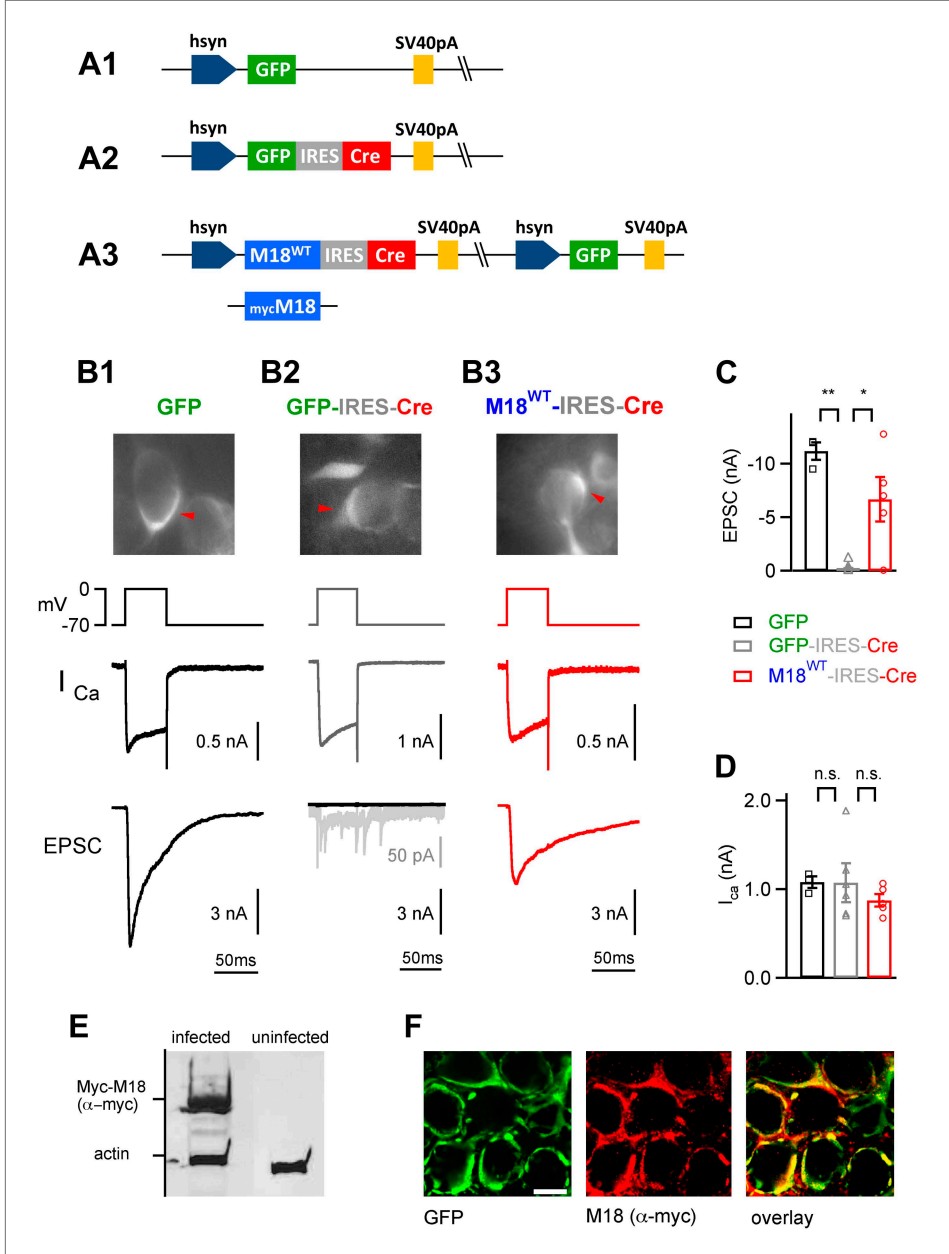

**Figure 3**. Endogenous floxed Munc18-1 can be removed and replaced by recombinant protein in vivo at the calyx of Held. (**A**) Scheme of the adenoviral DNA constructs. **A1**: control vector driving the expression of GFP alone; **A2**: control vector driving the expression of GFP-IRES-Cre; **A3**: three-protein expression vector, which drives the expression of Munc18-1 (either wild-type or myc-tagged), Cre-recombinase, and GFP from an additional cassette in the viral genomic backbone (see 'Materials and methods'). (**B**) Results from paired recordings from calyx synapses in *Munc18-1lox/lox* mice expressing either GFP alone (**B1**), GFP-IRES-Cre (**B2**), or M18WT-IRES-Cre and GFP (**B3**). Shown are the corresponding GFP-positive calyces (*top*), the presynaptic voltage-clamp protocols and presynaptic Ca²⁺ currents (*middle*), and the resulting postsynaptic EPSCs (*bottom*). Note the abolishment of release when Cre recombinase is expressed alone (**B2** – note remaining quantal release in grey trace with enhanced scale), and the rescue of release when Cre-recombinase is expressed together with M18WT protein (**B3**). These measurements were done following virus injection at P1, which we found necessary for efficient elimination of endogenous Munc18-1 protein in Cre expressing calyces. (**C** and **D**) Summary of EPSC amplitudes recorded in response to 50 ms presynaptic depolarization (**C**; see **B**), for the following conditions: expression of GFP alone (*left* bar, black data points), expression of Cre-recombinase (*middle*; grey data points), and expression of Cre-recombinase together with M18WT (*right*; red data points and see *Figure3—source data 1A*). The presynaptic Ca²⁺ current amplitudes were unaffected by
*Figure 3. Continued on next page*

*Figure 3. Continued*

genetic removal of Munc18-1 (**D** and ***Figure3—source data 1B***). (**E**) Adenovirus-mediated expression of myc-Munc18-1 in E2T packaging cells analyzed by SDS-PAGE and western blotting shows strong expression of recombinant protein (myc-Munc18-1, 67 kDa, actin, 43 kDa). (**F**) Immunohistochemistry of calyces of Held expressing the myc-tagged M18 construct in a P11 *Munc18^{lox/lox}* mouse after injection at P1. Antibodies against GFP (*left*, green channel) and c-myc (*middle*, red channel) were used; the overlay image is shown on the right. Scale bar, 10 μm.

The following source data and figure supplements are available for figure 3:

**Source data 1**.

**Figure supplement 1**. Early postnatal Cre expression was necessary for complete removal of endogenous Munc18-1.

**Figure supplement 2**. Early postnatal Cre expression was necessary for complete removal of endogenous Munc18-1: summary.

sufficient time for protein replacement (injection, P1; recordings 8–10 days later; see 'Materials and methods', ***Figure 3—figure supplements 1, 2***).

We first tested the feasibility of the Munc18-1 gene replacement strategy in a series of control experiments. When we expressed Cre recombinase in VCN neurons of *Munc18-1^{lox/lox}* mice using a GFP-IRES-Cre expression cassette (***Figure 3A2***), depolarization-evoked release at the calyx of Held synapse was essentially abolished (***Figure 3B2***). In contrast, control experiments with a GFP expressing virus (***Figure 3A1***) in *Munc18-1^{lox/lox}* mice showed normal depolarization-evoked EPSCs (***Figure 3B1***). On average, EPSCs were 0.251 ± 0.199 nA (n = 6 cells) and 11.2 ± 1.0 nA (n = 3 cells) when presynaptic neurons expressed GFP-IRES-Cre and GFP respectively (***Figure 3C***; p<0.01). The amplitude of presynaptic $Ca^{2+}$ currents was unchanged under both conditions (1.07 ± 0.20 and 1.08 ± 0.08 nA for GFP-IRES-Cre and GFP respectively; see also ***Figure 3D***). Thus, Cre recombinase was active, and 8–10 days following virus injection, all functionally relevant copies of Munc18-1 proteins had disappeared from calyx terminals. Additional experiments showed that virus injections early postnatally at P1 were necessary for complete removal of Munc18-1 function (***Figure 3—Figure supplements 1, 2***).

We next wished to test whether the simultaneous re-expression of Munc18-1 protein, together with the expression of Cre recombinase and GFP in the three-protein expression construct (***Figure 3A3***), would lead to efficient rescue of the Munc18-1 k.o. release phenotype. Indeed, depolarization-evoked release responses were nearly completely rescued (6.7 ± 2.3 nA, n = 5; ***Figure 3B3,C***). In all cases, presynaptic $Ca^{2+}$ currents were unchanged (***Figure 3B,D***; p>0.05), which suggests that Munc18-1 is necessary for vesicle fusion, but not for presynaptic $Ca^{2+}$ channel function.

In a third line of control experiments, we used a Myc-tag labeled Munc18-1 construct in the three protein expression construct (***Figure 3A3***, bottom). Western blotting and immunohistochemistry with an anti-Myc antibody showed that recombinant Munc18-1 protein was produced in cell lines (***Figure 3E***), and transported to the calyx of Held nerve terminals following in vivo expression in the VCN (***Figure 3F***). Together, these experiments suggest that simultaneous expression of Cre-recombinase, and a rescue Munc18-1 construct in VCN neurons of *Munc18-1^{lox/lox}* mice leads to the exchange of the endogenous Munc18-1 by the recombinant protein.

## PKC phosphorylation sites in Munc18-1 are essential for PTP

Having established an in vivo gene replacement approach for Munc18-1, we next studied the role of PKC phosphorylation of Munc18-1 during presynaptic plasticity, by replacing the endogenous Munc18-1 with a PKC-insensitive mutant. The PKC-insensitive Munc18-1 mutant carried alanine substitutions at the two PKC consensus sites (S306A, S313A) as well as at a third serine residue (S312A; called M18^{SA} mutant here; ***Wierda et al., 2007***). As a control, we used the M18^{WT} construct (***Figure 4A***). Following injection of either one of the two protein constructs into the VCN of *Munc18-1^{lox/lox}* mice at P1, EPSCs and presynaptic plasticity were measured in slices made 8–10 days later. The mutant and wild-type forms of Munc18-1 produced baseline EPSCs of similar amplitudes (2.87 ± 0.59 nA and 2.91 ± 0.61 nA respectively; p=0.96; ***Figure 4B1–D1,E***). However, when we applied brief high-frequency trains of afferent fiber stimulation (100 Hz, 4 s), we observed a striking difference in presynaptic plasticity

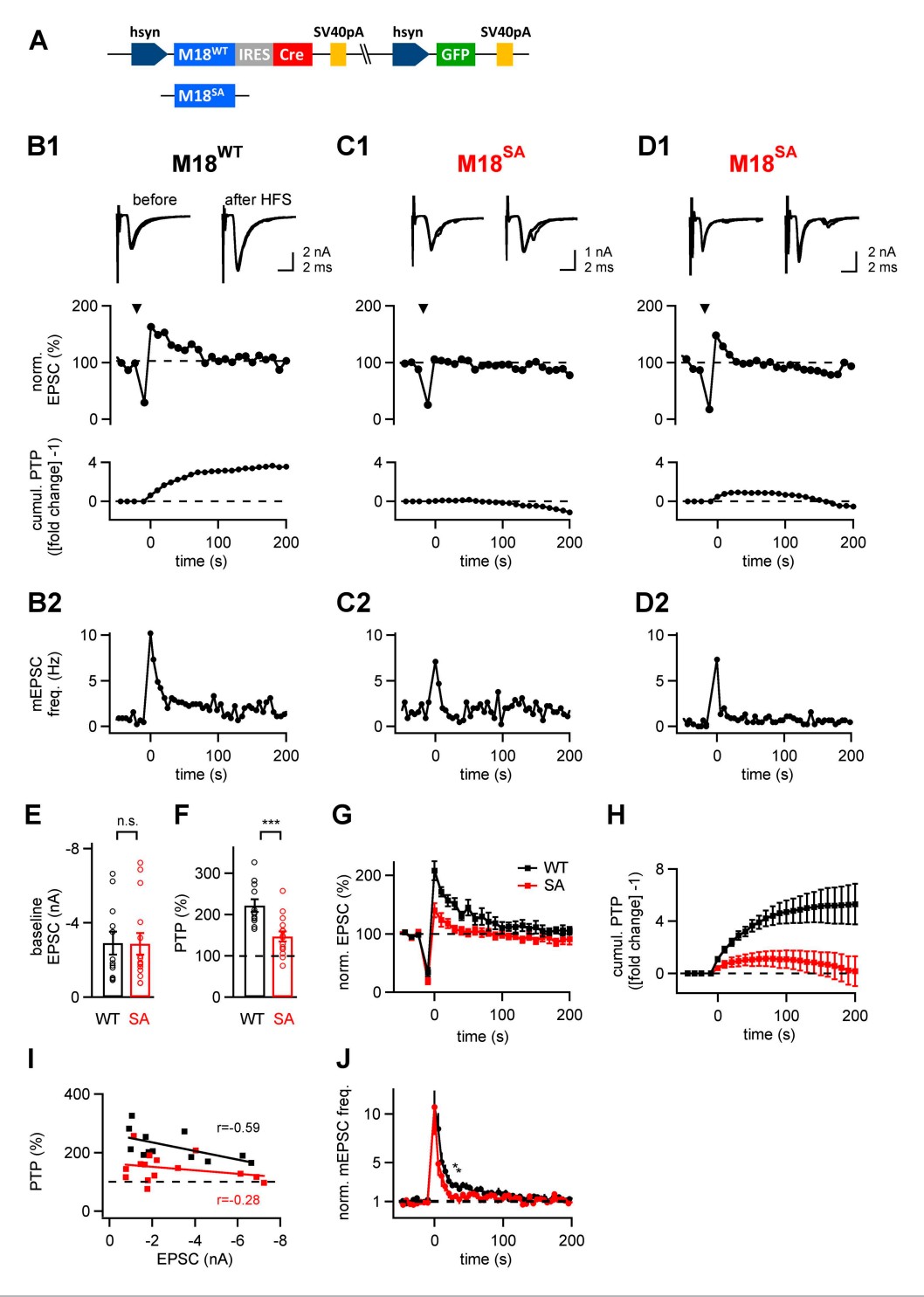

**Figure 4**. The PKC phosphorylation sites of Munc18-1 are necessary for the expression of post-tetanic potentiation, PTP. (**A**) Scheme of the three-protein expression constructs used for the experiments shown in this figure. Either wild-type Munc18-1 (M18^WT) or the PKC-phosphorylation site triple mutation (S306A, S312A, S313A; M18^SA) were expressed together with Cre-recombinase and GFP. (**B-D**) PTP from three example cells, one rescued with wild-type Munc18-1 (M18^WT, **B1**, **B2**), the other two rescued with the PKC-phosphorylation site deficient mutant (M18^SA; **C1**, **C2** and **D1**, **D2**). From *top* to *bottom*, individual EPSC traces before (*left*) and after (*right*) PTP; plots of relative EPSC amplitudes vs time (PTP plot), and plots of cumulative PTP vs time.
*Figure 4. Continued on next page*

*Figure 4. Continued*

Note that PTP was absent (**C1**) or smaller in synapses rescued with M18$^{SA}$; when substantial PTP remained, it decayed more rapidly (**D1**, *middle*). The panels in (**B2**–**D2**) plot the mEPSC frequency for the corresponding cells on the same time scale. (**E** and **F**) Summary plots of baseline EPSC amplitudes (**E** and *Figure 4—source data 1A*) and of peak PTP (**F** and *Figure 4—source data 1B*) for synapses rescued with wild-type Munc18-1 (M18$^{WT}$; *black* symbols) and with the mutant form (M18$^{SA}$; *red* symbols). (**G**) Average PTP plots for synapses rescued with M18$^{WT}$ (*black* symbols) and M18$^{SA}$ mutant (*red*; n = 12 and n = 15 cells, respectively). (**H**) Average cumulative PTP for synapses rescued with M18$^{WT}$ (*black* symbols) and M18$^{SA}$ mutant (*red*; n = 12 and n = 13 cells, respectively). (**I**) Plot of peak PTP amplitudes vs the basal EPSC amplitudes shows that PTP was reduced over the entire range of basal EPSC amplitudes. The correlation coefficients (r) are indicated. (**J**) Normalized average mEPSC frequency following PTP induction trains, recorded at synapses rescued with M18$^{WT}$ (black) and Munc18$^{SA}$ (red). In the range of 0–60 s, significance was tested by paired t-test. Two data points at around 30–40 s were found to be significantly different between the two conditions (p<0.05; see star symbols). Thus, Munc18-1 phosphorylation was not necessary for late release in the first 10 s interval after PTP induction, but probably supported some late enhanced mEPSC frequency, in agreement with the calyculin data in *Figure 1E*.

The following source data and figure supplements are available for figure 4:

**Source data 1**.

**Figure supplement 1**. mEPSC amplitude before and after PTP induction is unchanged in rescue experiments with wild-type or PKC-insensitive Munc18-1 mutant.

**Figure supplement 2**. PTP is blocked in synapses rescued by a phosphomimetic Munc18-1 mutant.

---

between synapses rescued with M18$^{WT}$, and with the M18$^{SA}$ mutant. While M18$^{WT}$-synapses showed normal PTP of 221 ± 15% of baseline EPSC amplitude (n = 12; see *Figure 4B1* for an example), synapses rescued with the Munc18$^{SA}$ mutant showed significantly smaller PTP (147 ± 13% of baseline; n = 15; p<0.001; *Figure 4B1–D1,F*). In some cases, PTP was abolished completely (*Figure 4C1*), whereas in other cases PTP was smaller and lasted for shorter times (*Figure 4D1*). To analyze the total amount of potentiation independent of its duration, we calculated the cumulative amount of PTP (*Figure 4B1–D1* bottom; *Figure 4H*). The cumulative PTP attained at 150 s was 5.06 ± 1.17 and 0.83 ± 0.85 (unit, fold change −1) in M18$^{WT}$ and M18$^{SA}$-rescued synapses, respectively (n = 12 and 13, respectively; p<0.01). A plot of the peak PTP amplitude vs baseline EPSC amplitude showed that PTP was reduced over the entire range of baseline EPSC amplitudes in the sample (*Figure 4I*).

Previous studies showed that following high-frequency stimulation to induce PTP, the mEPSC frequency is increased and this post-tetanic late release correlated with elevated residual $[Ca^{2+}]_i$ in the nerve terminal (*Habets and Borst, 2005*; *Korogod et al., 2005*). We analyzed the late post-tetanic release (*Figure 4B2–D2*) to investigate a possible role of Munc18-1 phosphorylation in this component of release. During the first 10 s interval following PTP induction trains, a strongly increased mEPSC frequency was observed both in synapses rescued with M18$^{WT}$ and M18$^{SA}$ (*Figure 4B2–D2*). This post-tetanic late release seemed to decay back to baseline faster in M18$^{SA}$ synapses than in M18$^{WT}$ synapses (*Figure 4B2–D2*). Indeed, we found that for a few sample points at intermediate times (at around 20–30 s following the PTP induction train), asynchronous release was reduced in M18$^{SA}$ synapses as compared to M18$^{WT}$ (*Figure 4J*, star symbols; p<0.05). In addition, the mEPSC amplitudes did not show a significant increase following PTP induction (*Figure 4—figure supplement 1*), which shows again that PTP under these stimulation conditions largely represents an increase in the amount of released quanta.

Taken together, replacing endogenous Munc18-1 protein by a PKC-deficient mutant selectively impaired the potentiation of evoked release during PTP, whereas the baseline transmitter release was unaffected, and late asynchronous release depended only marginally on PKC phosphorylation of Munc18-1. These experiments show that Munc18-1 is an important target protein for PKC phosphorylation during PTP.

## PKC phosphorylation of Munc18-1 is required for part of the phorbol ester potentiation

We next wished to test whether the PKC phosphorylation sites of Munc18-1 were also important for the phorbol ester-mediated potentiation of evoked and spontaneous release. The phorbol ester PDBu

(1 µM) still potentiated EPSCs in calyx synapses rescued with M18[SA], but the potentiation was significantly, about twofold smaller than with Munc18[WT] (157 ± 10% of control and 202 ± 18% of control, respectively; p<0.05; *Figure 5A–D*). Similarly, the frequency of spontaneous mEPSCs was potentiated by phorbol ester, but the potentiation was again smaller in synapses rescued with M18[SA] as compared to synapses expressing M18[WT] (307 ± 41% vs 518 ± 50%, respectively; p<0.05, *Figure 5H*). At first inspection, the baseline mEPSC frequency seemed to be higher in synapses rescued with M18[SA] as compared to M18[WT] synapses (*Figure 5G*). However, the difference did not reach statistical significance (p=0.50), and was caused by an outlier in the M18[SA] data set (*Figure 5G*, pink data point). When removing this data point, the average baseline mEPSC frequency was unchanged between M18[WT] and M18[SA] synapses (0.65 ± 0.22 and 0.48 ± 0.19 Hz, respectively; *Figure 5G*; p=0.54). Therefore, the reduced potentiation of mEPSC frequency by phorbol ester in M18[SA] synapses was not the result of a higher baseline mEPSC frequency (*Figure 5G*). We conclude that PKC phosphorylation of Munc18-1 accounts for part of the potentiation of evoked- and spontaneous release by phorbol esters.

## Discussion

Using a novel in vivo gene replacement strategy at a large CNS synapse, the calyx of Held, we show that PKC phosphorylation of the presynaptic protein Munc18-1 is critically important for PTP, a form of short-term enhancement observed at many CNS synapses. This identifies Munc18-1 as a presynaptic PKC target during PTP. Pharmacological experiments suggested that the PKC action during PTP is dynamic, since blocking phosphatases led to a marked prolongation of PTP (*Figure 1*). Together, these results show that a dynamic phosphorylation/de-phosphorylation cycle of Munc18-1, initiated by PKC activity, causes PTP.

A membrane-permeable peptide inhibitor of conventional PKC isoforms α and β suppressed PTP, whereas a similar PKA inhibitor had no effect (*Figure 2*). The peptide inhibitors have a more defined PKC isoform specificity than the synthetic PKC inhibitors used earlier (*Brager et al., 2003*; *Korogod et al., 2007*). This, together with recent findings from knock-out mouse lines (*Fioravante et al., 2011*), allows us to conclude that during PTP induction stimuli, conventional PKC isoforms are activated. A previous study evaluated the effects of BIS-like PKC inhibitors on PTP as non-specific (*Lee et al., 2008*). However, the genetic evidence for an involvement of conventional PKCs (*Fioravante et al., 2011*) and for the PKC phosphorylation sites of Munc18-1 in PTP (present study), as well as our pharmacological evidence with specific peptide inhibitors, firmly establishes the role of PKC and Munc18-1 phosphorylation in PTP. An earlier study found that phorbol esters translocates PKCε at the calyx of Held (*Saitoh et al., 2001*). Although we cannot exclude that PKCε contributes to phorbol ester potentiation of release, it seems unlikely that novel PKCs, including PKCε, play a major role during PTP, a conclusion which agrees with the inefficiency of PLC inhibitors (*Figure 2*). The inefficiency of PLC blockers in PTP further contrasts this form of short-term enhancement with augmentation, since the latter is sensitive to PLC blockers, but not affected by PKC inhibitors (*Rosenmund et al., 2002*).

We developed a novel gene replacement approach for the calyx of Held synapse, which makes use of a floxed mouse line, combined with virus-mediated expression of three proteins: (i) Cre-recombinase; (ii) the protein of interest in mutated, or wild-type form, and (iii) the reporter gene GFP. We used adenoviral vectors, which have a high cloning capacity and which allow the use of two expression cassettes (*Young and Neher, 2009*). The additional use of an IRES site allowed us to drive the expression of a third protein; the Cre-recombinase placed downstream of the IRES site was expressed efficiently as shown in control experiments (*Figure 3A2,B2*). Using this approach, we showed that replacing the endogenous (floxed) Munc18-1 with a Munc18-1 mutant resistant to PKC phosphorylation strongly suppressed PTP. These genetic experiments at the calyx of Held show that Munc18-1 is a necessary target for PKC during short-term enhancement of release.

Perturbing presynaptic PKC signaling with various methods, including PKC inhibition (*Figure 2*; see also *Korogod et al., 2007*), phosphatase block (*Figure 1*), and introducing a PKC insensitive Munc18-1 mutant (*Figure 4*), did not lead to changes of the baseline synaptic strength. This indicates that Munc18-1 is not phosphorylated in the nerve terminal under baseline conditions; a conclusion which agrees with previous work using a synaptosome preparation (*de Vries et al., 2000*). Therefore, it seems likely that during, or shortly after high-frequency activity, Munc18-1 is phosphorylated by a conventional PKC isoform, which induces short-term enhancement of release. About a minute later, a phosphatase de-phosphorylates Munc18-1, which terminates the enhanced evoked release during PTP. Thus, a dynamic regulation of the PKC phosphorylation state of Munc18-1 underlies the rise and decay of PTP.

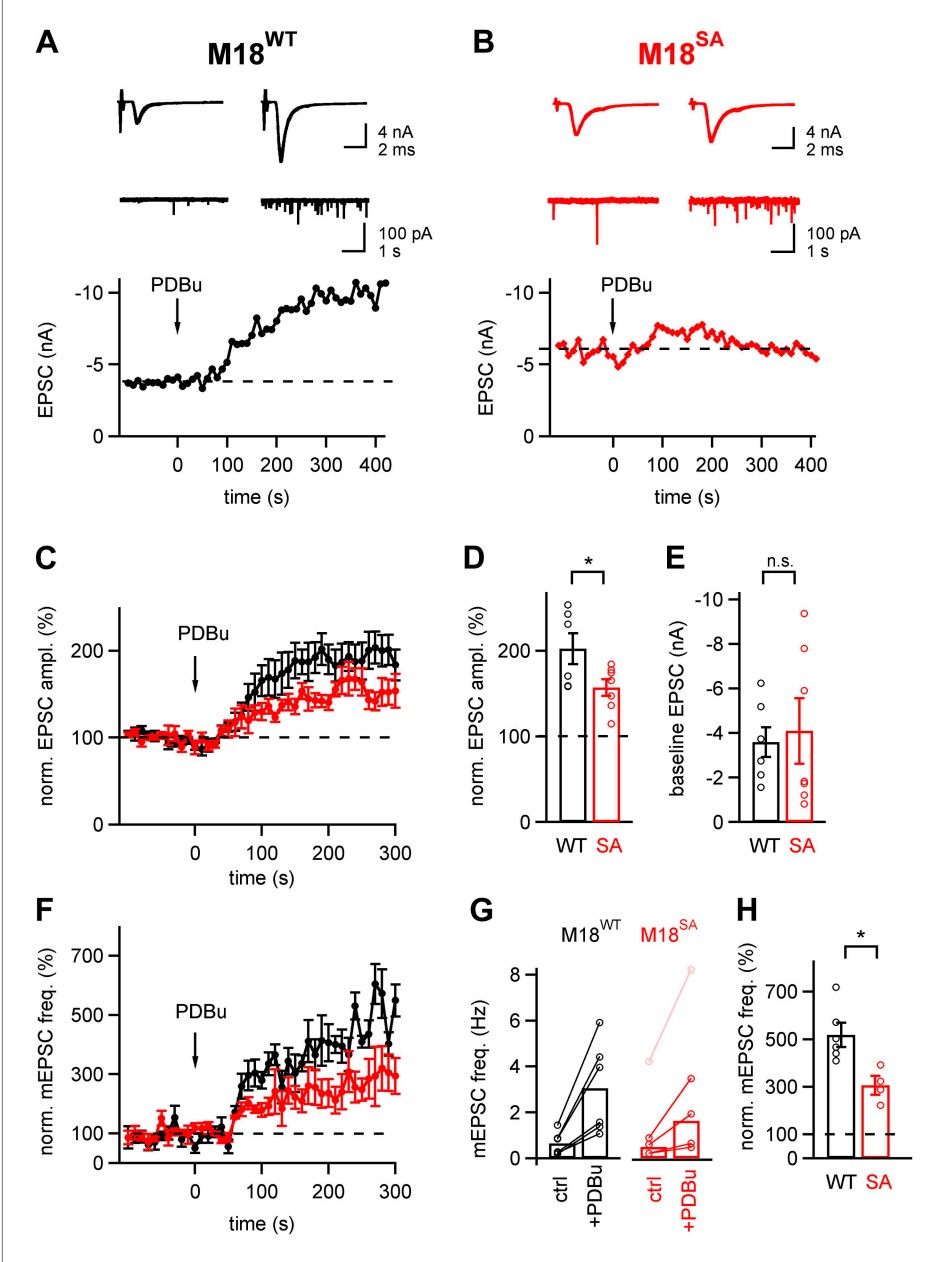

**Figure 5**. About half of the phorbol ester potentiation of evoked and spontaneous EPSC depends on PKC phospho-rylation of Munc18-1. (**A** and **B**) Evoked EPSC traces (*top*) and spontaneous EPSCs (*middle*) are shown both before (*left*) and after (*right*) application of 1 μM PDBu to the slice. The *bottom* panels show time plots of evoked EPSC amplitudes and their potentiation by PDBu. Data for a M18^WT rescued synapse (**A**) and for a M18^SA rescued synapse (**B**) are shown. (**C**) Average time courses of normalized EPSC amplitudes during PDBu potentiation for synapses rescued with M18^SA (*red* symbols) and M18^WT (black, n = 7 and n = 6 cells, respectively). (**D** and **E**) Quantifications of the average and individual values for EPSC potentiation (**D**, *Figure 5—source data 1A*) and for the baseline EPSC amplitudes (**E**, *Figure 5—source data 1B*) in synapses rescued with M18^SA (*red*) and with M18^WT (*black*). Note that ~half of the potentiation of evoked EPSC amplitudes depended on an intact Munc18-1 phosphorylation site. (**F**) Average time courses of normalized spontaneous EPSC frequency before and after PDBu application, both for synapses rescued with Munc18^SA (*red* symbols) and with M18^WT (*black*; n = 4 and 6, respectively). (**G**) Quantification of absolute mEPSC frequencies before and after PDBu application, for synapses rescued with M18^WT and M18^SA. For the M18^SA data, an outlier data point with an unusually high baseline frequency (4 Hz; pink symbols) was removed when calculating the average absolute mEPSC frequencies (see *Figure5—source data 1C*). (**H**) Average relative
*Figure 5. Continued on next page*

*Figure 5. Continued*

potentiation of mEPSC frequency under both conditions. Note that about half of the potentiation of spontaneous release depends on the PKC phosphorylation of Munc18-1 (see *Figure5—source data 1D*).

The following source data and figure supplements are available for figure 5:

**Source data 1**.

**Figure supplement 1**. Model of Munc18-1 PKC phosphorylation and de-phosphorylation and its effect on presynaptic plasticity.

---

Using the gene replacement approach, we also studied the potentiation of transmitter release by phorbol esters. We found that the Munc18-1 phosphorylation sites were necessary for ~half of the phorbol ester-mediated potentiation of evoked and spontaneous release (*Figure 5*). Phorbol esters, analogues of the membrane lipid product DAG, can activate several proteins including novel and conventional PKCs, Munc13-1, and other C1 domain containing proteins (*Newton, 2001*; *Brose and Rosenmund, 2002*; *Rhee et al., 2002*). Therefore, it is expected that phorbol ester activation extends beyond the conventional, $Ca^{2+}$-dependent PKCs that are activated during PTP (see Discussion above). Indeed, we have shown previously that ~half of the phorbol ester potentiation of spontaneous release at the calyx synapse depended on an intact DAG binding site of Munc13-1 (*Lou et al., 2008*). Thus, phorbol ester activates at least two parallel signaling pathways at the release machinery: on the one hand, Munc13-1 via direct binding to its C1 domain (*Betz et al., 1998*; *Rhee et al., 2002*; *Lou et al., 2008*), and on the other hand PKC which then phosphorylates Munc18-1 (*Figure 5*; *Wierda et al., 2007*; see *Figure 5—figure supplement 1*). The specificity of activation of conventional PKCs and phosphatases during PTP likely indicates the existence of signaling subcompartments in the presynaptic nerve terminal. Indeed, a recent proteomics study showed that various protein phosphatases and conventional PKCs are enriched at short distances to P/Q-type and N-type $Ca^{2+}$ channels (*Müller et al., 2010*), indicating a PKC–protein phosphatase signaling complex at the active zone.

How might phosphorylated Munc18-1 increase transmitter release during PTP? Munc18-1 was first described as a protein which tightly binds to Syntaxin (*Pevsner et al., 1994*; *Dulubova et al., 1999*). The binding of Munc18-1 to Syntaxin is thought to maintain the latter in a closed state not amenable to SNARE complex formation. An early study showed that the phosphorylated Munc18-1 has a reduced binding affinity for Syntaxin (*Fujita et al., 1996*); thus, PKC phosphorylation could aid in the transition from a binary Munc18-1/Syntaxin complex to incorporation into SNARE complexes. However, it was shown later that Munc18-1 also binds to the SNARE complex (*Dulubova et al., 2007*), and stimulates SNARE-mediated vesicle fusion in a reconstituted system, by binding to the partially assembled SNARE complex (*Shen et al., 2007*). Therefore, Munc18-1 likely remains present at the partially formed SNARE complexes of docked and readily releasable vesicles; interestingly, Munc13-1 is also present in this complex (*Ma et al., 2013*). In this model, PKC phosphorylation of Munc18-1 would then increase the release probability of readily releasable vesicles, in what appears a 'post-priming' regulatory step of vesicle fusion (see schematic in *Figure 5—figure supplement 1*).

A 'post-priming' regulation of the fusion competence of docked vesicles during PTP is consistent with previous $Ca^{2+}$ uncaging data at the calyx synapse. It was shown that phorbol esters induce a shift and a decreased slope in the dose–response curve between transmitter release rate and presynaptic $Ca^{2+}$ concentration (*Lou et al., 2005*); this modulation was reduced by PKC inhibitors (*Korogod et al., 2007*). We could not apply $Ca^{2+}$ uncaging to study PTP, since PTP is not observed during presynaptic whole-cell recordings (*Korogod et al., 2005*; *Lee et al., 2010*). To gain further insights into the mechanism of release modulation following Munc18-1 phosphorylation, we made use of a *phosphomimetic* mutation of Munc18-1, in which the S306 and S313 sites were changed to aspartate residues (called M18^SD mutant). With this mutation, however, functionally rescued synapses were difficult to obtain, and synapses which were rescued showed significantly smaller EPSCs as compared to synapses rescued with M18^WT and M18^SA constructs ($p < 0.05$; *Figure 4—figure supplement 2*). The M18^SD mutant led to a block of PTP consistent with the role of PKC phosphorylation in PTP, but the paired-pulse ratio was not changed significantly (*Figure 4—figure supplement 2*). Thus, while phosphorylation of Munc18-1 is required for PTP, mimicking its phosphorylation is not sufficient to cause a constitutive increase in release probability. We cannot rule out alternative explanations, like homeostatic plasticity (*Davis, 2013*) which could have reverted

the increased release probability, or incorrect folding of the M18$^{SD}$ mutant protein; the latter might also be the reason for the lower rescue efficiency. Taken together, the M18$^{SD}$ mutant data do not allow us to further extend our model of how Munc18-1 phosphorylation by PKC leads to an increased transmitter release.

Munc18-1 is essential for vesicle fusion at synapses (*Figure 3*; *Verhage et al., 2000*), and Munc18-1 is expressed widely in the brain (*Website, 2012*; *Lein et al., 2007*). However, PTP is probably not observed at all synapses in the brain, and target-cell specific differences in PTP mechanisms have been observed for hippocampal mossy fiber synapses (*Lee et al., 2007*). A long-lasting residual Ca$^{2+}$ signal, caused by mitochondrial Ca$^{2+}$ uptake and release mechanisms, is important for short-term enhancement of release including PTP (*Kamiya and Zucker, 1994*; *Regehr et al., 1994*; *Tang and Zucker, 1997*; *Habets and Borst, 2005*; *Korogod et al., 2005*; *Lee et al., 2007*). It is possible that differences in Ca$^{2+}$ extrusion mechanisms, as well as differential subcellular localization of members of the signaling complex downstream of Ca$^{2+}$, like conventional PKCs, determine the expression of PTP at a given synaptic connection.

Together with previous work (*Wierda et al., 2007*), our study implies that besides having an essential function for vesicle fusion, Munc18-1 can also modulate the release process following PKC phosphorylation. Our study identifies an important physiological context, post-tetanic potentiation, for this presynaptic regulatory pathway. We hypothesize that Munc18-1 is present at the partially formed SNARE complex of readily releasable vesicles, where it would exert a modulatory role on the fusion process, depending on its phosphorylation state. Future studies could investigate the role of Munc18-1 dependent presynaptic plasticity for information processing in neuronal networks, by making use of the PKC-insensitive mutation in transgenic mouse approaches.

## Materials and methods

### Mouse breeding and stereotactic surgery

Protocols of animal experiments with mice and rats were approved by the Veterinary Office of the Canton of Vaud, Switzerland. For the experiments shown in *Figures 3–5*, *Figure 3—figure supplements 1, 2*, *Figure 4—figure supplements 1, 2*, we used floxed Munc18-1 mice (*Heeroma et al., 2004*) that were generated by insertion of loxP sites flanking exon 2 of the mouse *Stxbp1* gene by homologous recombination in embryonic stem cells (*Heeroma et al., 2004*). Floxed *Stxbp1* mice were bred to homozygosity and referred to as *Munc18-1$^{lox/lox}$* mice.

*Munc18-1$^{lox/lox}$* mice were injected at postnatal day 1 (P1) with adenovirus unilaterally into the ventral cochlear nucleus (VCN) under isoflurane anesthesia, following a subcutaneous lidocaine injection. In general, mice were used for brainstem slice preparation 8–10 days following virus injection. In some experiments with Cre-recombinase alone, stereotactic injections were performed at different ages (P1–P6) to test the efficiency of Munc18-1 removal depending on postnatal injection time. In these initial experiments, slices were made at 7–8 days following injection (*Figure 3—figure supplements 1, 2*).

The stereotactic coordinates were adjusted from the previously established VCN injections at P6 mice (*Kochubey and Schneggenburger, 2011*). Lambda was located through the skin, still transparent at P1, and the mouse head was aligned in a model 900 stereotactic instrument (Kopf Instruments, Tujunga, CA) for the lambda, and the point 3.7 mm anterior from lambda, being in one horizontal plane and lying on the longitudinal axis of the instrument. The skin and the skull were co-punctured at two points which were 0.3 and 0.9 mm posterior, and 1.57 mm lateral from lambda. The virus (0.2 µl per site) was injected with a 35G stainless steel needle (Coopers Needle Works, Birmingham, UK) inserted through the punctures, at three sites vertically spaced at 150 µm from each other (maximal depth, 4.0 mm from surface). The injection rate was 80 nl/min, using a SP120PZ syringe pump (WPI, Berlin, Germany) and a 10 µl syringe (Hamilton, Bonaduz, Switzerland). The mice recovered from anesthesia in 10–20 min and were brought back to the mother.

### In vivo gene replacement strategy of Munc18-1

In order to replace the endogenous Munc18-1 protein of *Munc18-1$^{lox/lox}$* mice, we developed adenovirus vectors which, for the final experiments, were capable to drive the expression of three proteins: Munc18-1 (either in wild-type, or mutated form); Cre-recombinase; and GFP to label calyces of successfully transduced presynaptic neurons in the VCN (see *Figure 3A1–A3*). The constructs were designed as follows. We used an open reading frame of a codon-optimized Cre recombinase (*Gradinaru et al., 2010*) which was placed downstream of an IRES sequence. The IRES sequence was preceded either by the GFP sequence (see *Figure 3A2*), or by the Munc18-1 sequence (splice variant b), in wild-type

or mutated form (*Figure 3A3*, *Figure 4A*). In the phosphorylation deficient mutant (M18$^{SA}$), three serine residues 306, 312 and 313 were changed to alanine (*Wierda et al., 2007*). In the phosphomimetic mutant, the serine residues S306 and S313 were changed to aspartate (called M18$^{SD}$ mutant; see *Figure 4—figure supplement 2*; 'Discussion'). All expression cassettes were under control of the neuron-specific human synapsin1 promoter (*Kügler et al., 2003*; *Young and Neher, 2009*), were preceded by the Kozak sequence GCCACC, and followed by the SV40 polyadenylation signal.

A second-generation serotype 5 adenovirus system was used to deliver and drive the expression of the constructs in vivo. A shuttle vector pDC511 (Microbix Biosystems, Ontario, Canada) encoding eGFP-IRES-CreT cassette was used in combination with a custom-modified pBHGfrtΔ1, 3FLP adenovirus backbone (Microbix Biosystems) in which the gene 2a was also deleted (*Zhou and Beaudet, 2000*; *Young and Neher, 2009*). To enable simultaneous eGFP expression in rescue experiments, the shuttle vectors encoding for Munc18$^{WT/SA}$-IRES-Cre cassettes were combined with a backbone carrying an additional hsyn:eGFP expression cassette (*Young and Neher, 2009*). Adenovirus was propagated and purified as previously described (*Kochubey and Schneggenburger, 2011*) using the E2T packaging cell line (*Zhou and Beaudet, 2000*), including the plaque purification step. Final purification was done from the total lysate of 5 × 15 cm cell culture plates using Adenopack 100 RT kit (Sartorius, Aubagne, France) with the final buffer containing (in mM) 250 sucrose, 10 HEPES, 1 MgCl$_2$, pH 7.4, which typically resulted in ~750 μl of injection-ready virus stock with $1$–$2\cdot10^{12}$ particles/ml titer (OD$_{260}$).

## Slice electrophysiology

For the experiments shown in *Figures 1 and 2*, transverse slices of brainstem (200 μm) containing the medial nucleus of the trapezoid body (MNTB) were prepared from Wistar rats at P8–P10. For the experiments in *Figures 3–5*, *Munc18-1$^{lox/lox}$* mice that had undergone stereotaxic surgery at P1, to express one of the above described expression constructs in the VCN, were used at P9–P12. We used the eGFP-fluorescence of calyces of Held to select for presynaptically transduced neurons using a monochromator (TILL Photonics; Gräfelfing, Germany) at an excitation wavelength of 470 nm, a filter set with a 470/30 bandpass excitation filter, Q495LP dichroic mirror, and dual-band eGFP+IR emission filters (AHF Analysentechnik, Tübingen, Germany). Whole-cell patch-clamp recordings from MNTB neurons, or from pairs of calyces of Held and MNTB neurons were made at room temperature (21–24°C) using an EPC-9/2 double patch-clamp amplifier (HEKA Elektronik, Lambrecht, Germany). The microscope set-up was a BX-51WI upright microscope (Olympus, Tokyo, Japan), equipped with a 60×/0.9 NA water-immersion objective (Olympus), IR-DIC illumination system (Olympus) and an Andor iXon 885 EM-CCD camera (Andor Technology, Belfast, UK). In case of fiber stimulation experiments, afferent axonal fibers were stimulated with a custom-made platinum-iridium bipolar electrode, which was placed close to the midline of the brainstem slice. In these cases, MNTB neurons that were amenable to successful midline stimulation were pre-selected, by measuring the action current generated at the calyx of Held synapse in response to afferent stimulation (*Borst et al., 1995*). It is possible that by this procedure, we also selected for calyces of Held in which Munc18-1 function was rescued successfully, since calyces with little rescue might not respond to the afferent fiber test. During whole-cell recordings of postsynaptic MNTB neurons, the series resistance (R$_s$; 3–10 MΩ) was compensated by up to 90%; the remaining R$_s$ error in postsynaptic currents was corrected offline (*Meyer et al., 2001*). In presynaptic recordings, R$_s$ was 8–20 MΩ and was compensated by ~ 50%.

## Solutions and drug application

The slice incubation solution contained (in mM): 125 NaCl, 25 NaHCO$_3$, 2.5 KCl, 1.25 NaH$_2$PO4, 1 MgCl$_2$, 2 CaCl$_2$, 25 glucose, 0.4 ascorbic acid, 3 myo-inositol, 2 Na-pyruvate, pH 7.4 when bubbled with 95% O$_2$/5% CO$_2$ at 37°C. The solution for slice patch-clamp recording was similar, but contained in addition 2 μM strychnine and 10 μM bicuculline for the fiber stimulation experiments (*Figures 1, 2, 4 and 5*; *Figure 2—figure supplements 1, 2*; *Figure 4—figure supplements 1, 2*). For the paired pre- and post-synaptic recordings (*Figure 3*, *Figure 3—figure supplements 1, 2*), this solution contained, in addition, 10 mM tetraethylammonium chloride (TEA-Cl), 1 μM tetrodotoxin (TTX), 50 μM D-2-Amino-5-phosphonopentanoic acid (D-AP5), 2 mM γ-D-glutamylglycine (γ-DGG), 100 μM cyclothiazide (CTZ). The patch-pipette solutions contained (in mM): 140 Cs-gluconate, 20 TEA-Cl, 10 HEPES, 5 Na$_2$Phosphocreatine, 4 Mg$_2$ATP, 0.3 Na$_2$GTP, pH 7.2. This solution was supplemented with 5 or 0.1 mM Cs-EGTA for post- or presynaptic recordings, respectively. TTX, D-AP5, CTZ and γ-DGG

were from BIOTREND (Wangen, Switzerland). All the other chemicals were from Sigma Aldrich/Fluka (Buchs, Switzerland) unless indicated.

N-myristoylated cell-permeable PKC and PKA inhibitor peptides with specific pseudosubstrate sequences (myr-FARKGALRQ-amide, and myr-GRTGRRNAI-amide, respectively) were purchased from Merck (Darmstadt, Germany), reconstituted in $H_2O$ at 10 mM, and kept as frozen stocks. For the experiments, one slice at a time was pre-incubated in the recording chamber by circulating 10 ml of extracellular solution containing 10 µM of inhibitor peptide for 30 min at 32–36°C as set by a temperature control unit (Warner Instruments, Hamden, CT). Thereafter, the perfusion solution was allowed to cool to room temperature, and fiber-stimulation experiments were done in the continuous presence of the inhibitor peptide. Calyculin A (Millipore, Zug, Switzerland; 1 µM final concentration), phorbol-12,13-dibutyrate (Merck; 1 µM) and Neomycin (Sigma; 10 µM) were acutely applied via a gravity-driven bath perfusion. U73122 (Merck) was dissolved in DMSO (Sigma), and slices were incubated at the final concentration of 3 µM U73122, or in 0.1% DMSO vehicle for at least 45 min prior to electrophysiology experiments, using a small slice keeping chamber containing ~20 ml of $O_2/CO_2$ equilibrated keeping solution (see above).

### Immunohistochemistry and western blot analysis

For analysis of Munc18–1 expression in calyces of Held, mice were deeply anesthetized 8–10 days after virus injection by pentobarbital (Streuli, Uznach, Switzerland; 100–200 mg/kg peritoneally), and transcardially perfused with 4% paraformaldehyde (PFA) in phosphate buffered saline (PBS). After post-fixation in 4% PFA and dehydration in 30% sucrose in PBS, transverse brainstem sections containing the MNTB were cut frozen at 40 µm on a sliding microtome Hyrax S30 (Carl Zeiss, Oberkochen, Germany). The sections were processed as described (*Felmy and Schneggenburger, 2004*). We used chicken anti-GFP (1:1000; 13,970; Abcam, Cambridge, UK) and mouse anti-myc antibodies (1:50; 9B11; Cell Signaling Technology, Boston, MA), and secondary Alexa488 goat anti-chicken (A11039) and Alexa647 donkey anti-mouse (A31571) antibodies (from Life Technologies, Carlsbad, CA), which were applied overnight at 4°C, at 1:200 dilutions. The sections were mounted in DAKO fluorescence mounting medium (Dako, Glostrup, Denmark) and imaged on an inverted SP2 confocal microscope (Leica Microsystems) with a 40× oil immersion objective using 488 and 633 nm laser lines.

### Data analysis

Data were analyzed using custom-written routines in IgorPro 6.2 (Wavemetrics, Lake Oswego, OR). Spontaneous mEPSCs were detected by a semi-automated analysis routine in IgorPro, implementing template-matching detection algorithm (*Clements and Bekkers, 1997*). Detected mEPSC events were visually inspected before acceptance. The mEPSC frequency time courses were constructed by splitting the 9.7 s long mEPSC traces in two time intervals (*Figures 1E, 4B2–D2*) or by using the entire interval as one time bin (*Figure 5F*). Statistical significance was assessed by unpaired two-tailed Student's *t* test. When appropriate, a paired Student's *t* test was employed (*Figure 1B,C,F,G, 5G, Figure 2—figure supplements 1, 2, Figure 4—figure supplement 1*). Error bars report the standard error of the mean (SEM), and the statistical significance is indicated as follows: n.s., not significant ($p \geq 0.05$); *$p < 0.05$; **$p < 0.01$; ***$p < 0.001$.

## Acknowledgements

We thank Heather Murray for expert technical assistance and for help with molecular biology.

## Additional information

### Funding

| Funder | Grant reference number | Author |
| --- | --- | --- |
| Swiss National Science Foundation | 31003A_122496 | Ralf Schneggenburger |
| Swiss National Science Foundation | 31003A_138320/1 | Ralf Schneggenburger |

The funder had no role in study design, data collection and interpretation, or the decision to submit the work for publication.

## Author contributions

ÖG, Conception and design, Acquisition of data, Analysis and interpretation of data, Drafting or revising the article; OK, Conception and design, Analysis and interpretation of data, Drafting or revising the article, Contributed unpublished essential data or reagents; RFT, Contributed transgenic mouse model, Analysis and interpretation of data; MV, Contributed transgenic mouse model, Conception and design, Analysis and interpretation of data; RS, Conception and design, Analysis and interpretation of data, Drafting or revising the article

## Ethics

Animal experimentation: Protocols of animal experiments with mice and rats were approved by the Veterinary Office of the Canton of Vaud, Switzerland (authorizations 1880.2 and 1864.3). All surgery on mouse and rat pups was performed under isoflurane anesthesia and using local analgesia, and every effort was made to minimize suffering.

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
