## [Decision Letter]

Thank you for sending your work entitled “Munc18-1 is a dynamically regulated PKC target during short-term enhancement of transmitter release” for consideration at *eLife*. Your article has been favorably evaluated by a Senior editor and 2 reviewers, one of whom is a member of our Board of Reviewing Editors.

The Reviewing editor and the other reviewer discussed their comments before we reached this decision, and the Reviewing editor has assembled the following comments to help you prepare a revised submission.

1) Both reviewers found that your work is of high interest and novel. It advances our understanding on the mechanisms of PTP, and identifies in Munc18 a downstream target of PKC in generating PTP. Both reviewers however also felt that knowledge about the effects of a phosphomimetic mutant would further strengthen your conclusion, and in addition being useful to distinguish whether phosphorylation of Munc18 is not only required but also sufficient to trigger the high release probability state. While we felt that additional experiments in this direction would be very desirable, they may take considerable additional time that would prevent you from timely publication. We would therefore leave it up to you whether you want to add these data that you may already have obtained, or discuss the limitations that are related to the lack of data on the phosphomimetic mutants.

2) You also compared calcium currents and found no significant difference when Munc18 was intact, removed or rescued in the Calyx of Held in Figure 2. While the unaltered EPSC amplitude suggests that, the result should be displayed, given that calcium current triggers transmitter release.

3) PTP was induced by brief pulses at 100 Hz for 4 s. This stimulus induces ∼15% increase in the mEPSC amplitude in rat and mouse calyx, as reported by Wu's group (2009) and Regehr's group (2011). Consequently, compound fusion is suggested to contribute to PTP. Please report the mEPSC amplitude in every group of data in this manuscript, and discuss your results in case they differ to previously published work in context of putative mechanisms (e.g., compound fusion).

---

## [Author Response]

1) We would like to thank the editors and reviewers for their appreciation of our study. We have added the results of a phosphomimetic Munc18-1 mutation, in which the two PKC phosphorylation sites (Munc18-1 S306, S313) were changed to aspartate (called M18SD mutant). While the phosphomimetic mutation again interfered with PTP, it did not produce an increased release probability, as measured by paired-pulse ratio. This data is now added as Figure 4—figure supplement 2, and discussed in the text. While this data could indicate that Munc18-1 phosphorylation is necessary, but *not sufficient* to induce a high release probability state, we also observed that the rescue efficiency with this mutant was much lower than with the M18SA mutant, or the M18WT form. Therefore, we would like to restrain from drawing additional mechanistic conclusions based on this mutation, as discussed in more detail in the paper.

2) We think the Ca^2+^ current results were already displayed in the previous version (Figure 2, and Figure 2, now the same subpanels of Figure 3). In addition, we now report the average values of Ca^2+^ currents in the results text.

3) As requested, we now analyzed all PTP data sets for a possible increase of the mEPSC amplitude following the PTP induction trains. Except for the very first data set (control group of Figure 1), the increase in mEPSC amplitudes following PTP induction trains did not reach statistical significance in all data sets of our study (Figure 2—figure supplement 1 and Figure 4—figure supplement 1). Thus, while there might be a tendency for an increased mEPSC amplitude, this was not substantial with PTP induction trains of 100 Hz 4 s durations. This conclusion agrees with our original paper ([27]; see Figure 1), and with the paper by Xue et al. (2010) showing that a substantial increase in mEPSC amplitude, probably caused by compound fusion (19), occurs primarily with stronger PTP induction protocols (e.g., 100 Hz for 10 s).

We added the analysis of mEPSC frequency and mEPSC amplitude before and after PTP, under control conditions and in the presence of calyculin (new panels of Figure 1). This analysis also reveals a persistent increase in the mEPSC frequency after PTP in the presence of calyculin (Figure 1). We moved the data previously contained in Figure 1 (effect of the PKCi inhibitory peptide) to the new Figure 2, which now also displays the results with the phospholipase C blocker U73122 (previously shown as a supplementary v). We think that these changes have significantly improved the paper.